# Drug Metabolism of Hepatocyte-like Organoids and Their Applicability in In Vitro Toxicity Testing

**DOI:** 10.3390/molecules28020621

**Published:** 2023-01-07

**Authors:** Manon C. Bouwmeester, Yu Tao, Susana Proença, Frank G. van Steenbeek, Roos-Anne Samsom, Sandra M. Nijmeijer, Theo Sinnige, Luc J. W. van der Laan, Juliette Legler, Kerstin Schneeberger, Nynke I. Kramer, Bart Spee

**Affiliations:** 1Department of Clinical Sciences, Faculty of Veterinary Medicine, Regenerative Medicine Center Utrecht, Utrecht University, 3584 CT Utrecht, The Netherlands; 2Division of Toxicology, Wageningen University, 6700 EA Wageningen, The Netherlands; 3Institute for Risk Assessment Sciences, Utrecht University, 3584 CM Utrecht, The Netherlands; 4Department of Cardiology, Division Heart & Lungs, University Medical Center Utrecht, 3508 GA Utrecht, The Netherlands; 5Department of Surgery, Erasmus MC Transplant Institute, University Medical Center Rotterdam, 3015 CN Rotterdam, The Netherlands

**Keywords:** drug-induced liver injury, hepatic in vitro model, hepatotoxicity, intrahepatic cholangiocyte organoids, hepatocyte-like cells

## Abstract

Emerging advances in the field of in vitro toxicity testing attempt to meet the need for reliable human-based safety assessment in drug development. Intrahepatic cholangiocyte organoids (ICOs) are described as a donor-derived in vitro model for disease modelling and regenerative medicine. Here, we explored the potential of hepatocyte-like ICOs (HL-ICOs) in in vitro toxicity testing by exploring the expression and activity of genes involved in drug metabolism, a key determinant in drug-induced toxicity, and the exposure of HL-ICOs to well-known hepatotoxicants. The current state of drug metabolism in HL-ICOs showed levels comparable to those of PHHs and HepaRGs for CYP3A4; however, other enzymes, such as CYP2B6 and CYP2D6, were expressed at lower levels. Additionally, EC50 values were determined in HL-ICOs for acetaminophen (24.0–26.8 mM), diclofenac (475.5–>500 µM), perhexiline (9.7–>31.5 µM), troglitazone (23.1–90.8 µM), and valproic acid (>10 mM). Exposure to the hepatotoxicants showed EC50s in HL-ICOs comparable to those in PHHs and HepaRGs; however, for acetaminophen exposure, HL-ICOs were less sensitive. Further elucidation of enzyme and transporter activity in drug metabolism in HL-ICOs and exposure to a more extensive compound set are needed to accurately define the potential of HL-ICOs in in vitro toxicity testing.

## 1. Introduction

Drug metabolism is a key determinant in drug-induced toxicity [1]. The liver plays a crucial role in drug metabolism and is, therefore, susceptible to drug-induced injury. Despite the implementation of novel human-based strategies in drug development such as in vitro and in silico pre-clinical testing [2,3], drug-induced liver injury (DILI) remains a major cause for discontinuation of drug development and the withdrawal of drugs from the market [4]. Gaining human-relevant mechanistic insights into DILI is essential to improve toxicity prediction and further minimize adverse drug reactions.

The metabolism of drugs in the liver is generally a two-step process. In phase I, polar functional groups are added or opened up so that phase II enzymes can conjugate the drug to facilitate the drug’s excretion. The cytochrome P450 (CYP) superfamily forms the most prominent and most studied family of phase I biotransformation enzymes [5]. CYP oxidation often leads to bioactivation and is associated with DILI [6]. Genetic polymorphisms in common human CYP isoforms, including CYP2B6, CYP2C9, and CYP2D6, are a common mechanism of adverse drug reactions requiring hospitalization [7,8]. Generally, phase II metabolism by uridine 5′-diphospho-glucuronosyltransferases (UDP-glucuronosyltransferases, UGTs), sulfotransferases (SULTs), and glutathione transferases (GSTs) is a detoxification process, counteracting the reactivity of intermediate metabolites [9]. Hepatic transporters, including superfamilies ATP-binding cassette (ABC) transporters and solute carrier (SLC) transporters, are involved in the excretion of drugs and their conjugated metabolites. Inhibition of efflux transporters leading to intracellular accumulation is another risk factor for drug–drug interaction and DILI [10]. The activities of these two phases of drug metabolism and functionality of hepatic transporters are crucial in hepatotoxicity, as they are responsible for the (de)activation and excretion of chemicals [11,12].

Significant interspecies and interindividual differences in the expression and function of drug metabolism enzymes and transporters hamper accurate prediction of pharmacokinetics in patients and hepatotoxic potency of new drugs, as this is traditionally performed in animal models [13,14,15]. Over the years, the development of non-animal alternatives evolved to generate human-based toxicity data as well as to replace, reduce and refine animal use (3Rs) in safety evaluations [16,17]. To perform reliable human-based toxicity screens or mechanistic studies into DILI pathways, hepatic human in vitro models need to express morphological and functional features, such as drug metabolism, similar to an in vivo situation [18].

Human-based hepatic in vitro models have been developed using a range of cell sources, where primary human hepatocytes (PHHs), human hepatic cancer cell lines, and human stem cell-derived hepatocyte-like cells are three main hepatic cell sources used in current models [19,20]. In an effort to increase the in vitro toxicity prediction by these models, different approaches to enhance the physiological relevance and thereby maintain or improve hepatic function are being developed [21,22]. PHHs are considered the gold standard in in vitro toxicity testing, as their phenotype is most comparable to hepatocytes in vivo, which can be maintained longer due to advances in the culture method [23]. Besides interspecies differences in drug metabolism that are covered by the use of human-based in vitro models, interindividual differences in drug metabolism and thus drug sensitivity are of particular interest [6]. PHHs can represent real human population variability; however, their availability is limited and expansion is very difficult [24,25,26]. Donor- derived hepatic cell models, such as induced pluripotent stem cells (iPSCs) or adult stem cells (ASCs), can reflect the heterogenous phenotype of the human population and have the potential to be expanded for high-throughput purposes [27].

Human intrahepatic cholangiocyte organoids (ICOs) are liver-derived ASCs that form hollow polarized 3D structures in vitro and, once differentiated towards the hepatic lineage, show an increased expression of hepatic markers such as albumin, CYP enzymes, and transporters [28]. The applicability of these hepatocyte-like ICOs (HL-ICOs) for disease modelling and regenerative medicine has been described [29,30,31,32,33]; however, the potential of HL-ICOs as a novel cell model for DILI still needs to be explored [34]. Here, we sought to explore the potential of HL-ICOs for in vitro toxicity testing compared to PHHs and the tumor-derived hepatic cell line HepaRG. We focused on the expression of genes involved in phase I and II drug metabolism and hepatic transporters and phase I enzyme and UGT activity. Additionally, we tested a set of known hepatotoxic compounds, namely acetaminophen, diclofenac, perhexiline, troglitazone, and valproic acid, to study cytotoxicity after exposure.

## 2. Results

Intrahepatic cholangiocyte organoids (ICOs) were cultured in conventional Matrigel droplets and differentiated towards the hepatic lineage. After differentiation, hepatocyte-like ICOs (HL-ICOs) formed polarized 3D structures with a hollow lumen with a submembranous positivity for F-actin (Figure 1). The hepatic differentiation status of HL-ICOs indicated an increase in hepatic markers including albumin and CYP3A4 (Appendix A), comparable to previous studies [28,31,35]. The hepatic cell line HepaRG was cultured in standard 2D monolayer, and PHHs were cultured as monolayer in a collagen I sandwich (Figure 1), both with an F-actin located on intercellular filaments.

### 2.1. Expression of Phase I and II Enzymes, and Hepatic Transporters

Gene expression levels of selected key genes involved in phase I and II drug metabolism and hepatic transporters were examined in liver tissue from two donors, primary human hepatocytes, differentiated HepaRG and ICOs (3 donors) in expansion condition (EM), and hepatic differentiation condition (DM day 5 and 12).

Expression levels in ICOs of most phase I enzymes including major cytochrome P450 enzymes, such as CYP1A2, CYP2B6, and CYP2D6, improved upon hepatocyte differentiation but showed low expression compared to PHHs and HepaRGs (Figure 2). Expression of major CYP enzymes CYP3A4 and CYP2C9 increased upon ICO differentiation, where expression levels at day 12 of differentiation were higher compared to HepaRGs. CYP1A1 expression levels also increased upon differentiation of ICOs, where expression levels at day 12 of differentiation were higher compared to PHHs and HepaRGs. Other upregulated genes upon hepatic differentiation compared to expansion condition were (among others) CYP2C19, CES1, FMO4, and FMO5 in phase I, UGT2B7, UGT2B11, SULT1C2 in phase II, and transporters ABCG2, ABCB1, and ABCB11 (Figure 2). Hepatic transporters ABCB1, ABCB11, and ABCB8 were more highly expressed in HL-ICOs compared to HepaRGs.

Intraindividual differences between the three ICO donors could be observed in (among others) phase I enzymes CYP2C8, CYP2C19, CYP3A5, and CES1, phase II enzymes SULT1B1, UGT2B15, UGT2A1, SULT1A3, and transporters ABCB6, ABCG1, ABCB8, and SLCO2B1.

CYP family members were thoroughly studied to examine their role in DILI, especially the members most abundantly present in humans: CYP3A4, CYP2E1, CYP2C9, CYP2C8, and CYP1A2. Based on the increased expression of these CYP enzymes, differentiation day 12 was selected for further experiments. Due to practical considerations, four different ICO donors were used in further experiments.

### 2.2. Phase I and II Enzyme Activity

The activities of cytochrome P450 enzymes 1A2, 2B6, 2C9, 2D6, 2E1, and 3A4 and UGT were examined in the three hepatic cell models. Cells were exposed to two cocktails of, in total, seven specific enzyme substrates. Metabolite formation was used as a measure for activity of the specific CYP enzyme and UGT (Figure 3). For each cell model, metabolite formation was measured at three timepoints, which differed per hepatic cell model (HL-ICOs: 4, 8, 24 h; PHH: 1, 2, 4 h; HepaRG: 2, 4, 8 h). Metabolite formation rates were calculated using the linear correlation of formed metabolite (pmol/10^6^ cells) in time (Table 1).

Activity could not be determined in HL-ICOs for CYP2B6, CYP2C9, CYP2D6, and CYP2E1, as there was no metabolite formation. The chlorzoxazone metabolite (6-hydroxy-chlorzoxazone formed by CYP2E1) was also not formed by PHHs, and in HepaRG cells only in one (out of three) experiments at the last timepoint (8 h) of incubation, indicating low activity for CYP2E1 (data not shown). Metabolite formation by CYP2B6 (hydroxybupropion), CYP2C9 (4-hydroxytolbutamide), and CYP2D6 (dextrorphan) in both HepaRGs and PHHs showed a linear trend (Appendix A). In all three systems, depletion of bupropion was observed in control (no cells; data not shown), indicating degradation due to other components in the system, such as binding to the polystyrene culture plate.

CYP1A2 activity in HL-ICOs only showed metabolite (acetaminophen) formation in two of the four donors at the last timepoint (24 h) of incubation (Figure 3). CYP1A2 activity in HepaRGs was not consistent over the three independent experiments, as in one of the three experiments no metabolite was formed. CYP1A2 activity in PHHs was highest compared to the other two hepatic cell models. CYP3A4 activity in HL-ICOs showed intraindividual variation, as one of the four tested donors showed CYP3A4 activity comparable to PHHs (Figure 3; Table 1). CYP3A4 activity in the other three donors was comparable to HepaRGs. Glucuronidation of 7-hydroxycoumarin by UGT showed complete depletion of parent compound 7-hydroxycoumarin in all three hepatic models. In PHHs and HepaRGs, metabolite formation was to the same extent as the parent compound; however, in HL-ICOs, metabolite formation was only 24–72% of the parent compound (data not shown). UGT activity in HL-ICOs was lower compared to PHHs and HepaRGs and was variable between the different ICO donors (Figure 3; Table 1).

### 2.3. Cytotoxicity

HL-ICOs (four independent donors), PHHs, and the hepatic cell line HepaRG (three independent experiments) were exposed to five known hepatotoxic compounds for 48 h (Table 2; Figure 4). Concentration ranges differed per compound but were the same for the different hepatic cell models.

The determined EC50 of acetaminophen for the four ICO donors (24.01–24.87 mM) was higher compared to that for PHHs (4.19 mM) and the hepatic cell line HepaRG (ranging from 3.46 to 6.04 mM). For diclofenac, the EC50 was determined for only one of the tested ICO donors (475.5 µM), while for the other three ICO donors, the EC50 was higher than the highest tested concentration (500 µM). The EC50 of diclofenac for PHHs (421.2 µM) and HepaRGs (ranging from 272.4 to 434.9 µM) was fairly similar. Perhexiline exposure showed no cytotoxicity in three ICO donors; in one donor, cytotoxicity was observed in the highest concentration (32 µM; EC50 of 9.675 µM). PHHs and HepaRGs had a similar cytotoxicity curve (PHH: 8.072 µM; HepaRG: ranging from 10.45 to 26.37 µM). The determined EC50 of troglitazone in HL-ICOs and HepaRGs was in the same range and followed a similar trend (HL-ICOs: ranging from 23.13–90.83 µM; HepaRG: ranging from 14.89 to 45.17 µM). The EC50 of valproic acid could not be determined in HL-ICOs (PHH: 9.88 mM; HepaRG: ranging from 4.17 to 6.07 mM).

## 3. Discussion

Emerging advances in the field of in vitro and in silico toxicity testing attempt to meet the need for reliable human-based safety assessment in drug development [36,37]. Well-established in vitro models, such as PHHs or human hepatic cancer cell lines, are used for high-throughput toxicity screens [38,39,40] and/or in studies of the mechanisms driving DILI [41]. Intrahepatic cholangiocyte organoids (ICOs) have been recently described as a donor-derived hepatic in vitro model with potential in disease modelling and regenerative medicine [29]. Here, we explored the potential of liver-derived hepatocyte-like ICOs (HL-ICOs) in in vitro toxicity testing by quantifying the expression and activity of genes involved in drug metabolism and exposure to well-known hepatotoxicants.

Drug metabolism is of particular interest due to (de)toxification of compounds in the liver by phase I or phase II enzymes and excretion of compounds by hepatic transporters [6,12]. Two timepoints of hepatic differentiation of ICOs were included in the RNAseq analysis, as hepatic markers are known to rise and fall asynchronously, resulting in no optimal differentiation day for all markers [35]. Based on increased expression levels of CYP enzymes and hepatic transporters on the late differentiation day (d12), further experiments were executed in this differentiation window. In ICOs, the gene expression of abundant CYP enzymes in human CYP2B6, CYP2C9, and CYP2D6 improved upon hepatic differentiation; however, expression was lower than in PHHs. This was reflected in the CYP activity results, even though different ICO donors were used. CYP3A4 expression, which is responsible for metabolism of most therapeutic categories [5], was increased upon differentiation to levels higher than in HepaRG, which was reflected in the CYP activity data. One donor reached a formation rate comparable to PHHs, indicating the interindividual differences in CYP expression [5]. Even though CYP2D6 and CYP2C9 are highly variable in the human population [8], the used ICO donors did not show activity of these enzymes, as metabolite formation was not measurable. Expression of phase II enzymes in ICOs was generally higher than that of phase I enzymes. Notably, we observed relatively high expression of phase II enzymes in HL-ICOs compared to PHHs and HepaRGs, such as UGT2B11, UGT2B15, SULT1C2, and SULT1B1, suggesting differential activity in phase II metabolism pathways, as was observed by 7-hydroxycoumarin metabolite formation [42].

In order to further elucidate the potential of HL-ICOs in in vitro toxicity testing, the sensitivity of HL-ICOs to five well-known hepatotoxicants with different mechanisms of action was examined [43]. The three cell models were exposed to acetaminophen, a classic example of intrinsic DILI due to its predictable and dose-dependent toxicity [44]. The formation of its toxic metabolite NAPQI, catalyzed by CYP2E1, CYP3A4, and CYP1A2, is known to cause subsequent glutathione depletion [44,45]. However, we did not see a donor difference regarding the ICO donor with high CYP3A4 activity. The established EC50 in HL-ICOs was five-fold higher than those in PHHs and HepaRGs (which were comparable to the literature [19]). This difference could possibly be due to different media compositions (high levels of glutathione increase NAPQI conjugation) or increased activity in the alternative glucuronidation and sulfation pathways, as previously mentioned [45,46,47]. Exposure to the non-steroidal anti-inflammatory drug (NSAID) diclofenac showed that the sensitivity of one ICO donor was comparable to that of HepaRGs and PHHs [19]. Although no metabolite formation was measurable for CYP2C9 activity in HL-ICOs, the diclofenac data suggested that this specific donor did have CYP2C9 activity, as this CYP enzyme is involved in the bioactivation of diclofenac [48]. Perhexiline is a coronary vasodilator that was withdrawn from the market due to hepatotoxicity and neurotoxicity. The exact mechanism of perhexiline toxicity has not yet been clarified; however, it is suggested that perhexiline hepatotoxicity is mostly caused by the parent drug and that CYP2D6 is involved in the detoxification [49,50]. The observed cytotoxicity for PHHs and HepaRGs and one ICO donor was comparable to the literature [49]. Notably, no cytotoxicity was observed in three ICO donors, while higher toxicity was expected in a system with low metabolism [50,51]. Troglitazone, a thiazolidinedione derivative, is known to cause parent compound toxicity, but its metabolites also cause toxicity, such as inhibition of hepatic transporter BSEP, resulting in intrahepatic cholestasis [52]. Established EC50s by troglitazone exposure were comparable to those reported in the literature [19], even though a different trend was observed for PHHs compared to HepaRGs and HL-ICOs. Slight interindividual differences in cytotoxicity were observed between the ICO donors; however, this could not be linked to CYP and UGT activity data in this study. In the literature, troglitazone cytotoxicity cannot be correlated to CYP activity either; however, sulfotransferases possibly play a role in its cytotoxicity [53,54]. Anticonvulsant valproic acid hepatotoxicity is mainly caused by its metabolites, resulting in drug-induced steatosis [55]. Toxic metabolite formation is catalyzed by CYP2C9 and CYP2B6, the activity of which could not be measured in HL-ICOs. For HL-ICOs, an EC50 could not be established; however, the trend seems comparable to that of PHHs.

To improve the comparison between different cell systems, it is essential to determine the concentration that is actually available to be taken up by the cells in the in vitro system. In the case of lipophilic compounds such as perhexiline, in vitro system components such as Matrigel or medium components can decrease this available concentration [56,57]. In the case of bupropion (CYP2B6 activity), abiotic degradation of the compound was observed, especially in the HL-ICOs. Determining the real in vitro dose facilitates reliable extrapolation of in vitro data towards in silico models [18,58].

The current state of drug metabolism in HL-ICOs showed comparable levels for, among others, CYP3A4. However, the expression and activity of other enzymes, such as CYP2B6 and CYP2D6, need improvement in HL-ICOs compared to PHHs and HepaRGs. The donor-derived origin of the HL-ICOs was clearly represented in the CYP3A4, CYP1A2, and UGT activity data and in gene expression levels, such as for CYP2C8, CES1, and SLCO2B1, which are known to have polymorphisms [8,59,60]. Genotyping ICOs of different donors would give more insight into the donor differences and enable the selection of a panel of donors with slow/fast metabolizers or specific polymorphisms associated with DILI [48]. Moreover, extending the set of donors could also provide insight into sex-specific drug responses [61]. A more mechanistic approach would help in further characterization of HL-ICOs as an in vitro toxicity model [41,62]. Recent papers explored the potential of HL-ICOs with a more mechanistic approach to study the applicability of HL-ICOs as an in vitro model for cholestasis [63] and phospholipidosis [31]. While bile production in ICOs was shown to be low compared to HepaRGs and PHHs, ICOs were more sensitive in drug-induced phospholipidosis screening compared to HepG2 cells. A unique feature of HL-ICOs is their polarization and relative expression of hepatic transporters compared to PHHs, suggesting possibilities for HL-ICOs in toxicity screens involving transporters that are important in toxicity prediction [64]. Further characterization of the functionality of HL-ICOs, for example, using a comprehensive phase II enzyme activity assay [65,66], and their predictive potential to compound toxicity using a more extensive set of test compounds and readouts in a high-throughput fashion [67,68], will illustrate their potential in in vitro toxicity testing.

## 4. Materials and Methods

### 4.1. Cell Culture

LiverPool cryoplateable hepatocytes (pool of 10 donors, mixed gender; BioIVT, Hicksville, NY, USA) were cultured in a collagen I (Sigma-Aldrich, St Louis, MO, USA) sandwich in INVITROGRO CP medium (BioIVT) complemented with the TORPEDO Antibiotic mix (BioIVT). Seeding density was 49,000 cells/well in a 96-well plate (cytotoxicity assay) or 350,000 cells/well in a 24-well plate (CYP activity assay), according to the manufacturer’s instructions. Exposure of the hepatocytes was started 48 h after seeding. For exposure assays, INVITROGRO HI medium (BioIVT) was used, as recommended.

Undifferentiated HepaRG cells were purchased from Biopredic International (Saint Grégoire, France). Cells were cultured (passage number between p18 and p28) in T75 flasks in culture medium consisting of William’s E medium (without phenol red; Thermo Fisher Scientific, Waltham, MA, USA) supplemented with 1% (*v*/*v*) penicillin-streptomycin (Thermo Fisher Scientific), 10% (*v*/*v*) fetal bovine serum (FBS; Thermo Fisher Scientific), 50 µM hydrocortisone 21-hemisuccinate (Sigma-Aldrich, St Louis, MO, USA), 2 mM GlutaMax (Thermo Fisher Scientific) and 5 µg/mL insulin (Sigma-Aldrich). For differentiation, cells were cultured for 7–10 days and upon confluence, the monolayer was switched to differentiation medium (culture medium supplemented with 1% (*v*/*v*) DMSO). After differentiation, cells were trypsinized using TrypLE (Thermo Fisher Scientific) and seeded at a density of 65,000 cells/well in a 96-well plate (cytotoxicity assay) or 130,000 cells/well in a 24-well plate (CYP activity assay). Cells were allowed to attach for 24 to 48 h. Before exposure, cells were washed with assay medium (culture medium without FBS), after which exposure was started.

Intrahepatic cholangiocyte organoids (ICOs) were isolated from healthy liver biopsies that were obtained during liver transplantation at the Erasmus Medical Center Rotterdam in accordance with the ethical standards of the institutional committee to use the tissue for research purposes (ethical approval number MEC 2014-060). The procedure was in accordance with the Helsinki Declaration of 1975, and informed consent in writing was obtained from each patient. Obtained human liver material was frozen down in Recovery Cell Freezing Medium for future experiments or used for organoid isolation directly. Organoid isolation is described by Schneeberger et al. 2018 [35]. In short, small pieces of tissue were enzymatically digested at 37 °C. The supernatant was collected every hour, and fresh enzyme-supplemented medium was added to the remaining tissue until only ducts and single cells were visible. Cells were washed with DMEM Glutamax (supplemented with 1% (*v*/*v*) FBS and 1% (*v*/*v*) P/S) and spun down at 453 g for 5 min.

The cell suspension was cultured in Matrigel™ (Corning, New York, NY, USA) droplets in expansion medium (EM) until organoids were formed, as previously described [28]. EM consisted of Advanced DMEM/F12 (Life Technologies) supplemented with 1% (*v*/*v*) penicillin-streptomycin (Life Technologies), 1% (*v*/*v*) GlutaMax (Life Technologies), 10 mM HEPES (4-(2-hydroxyethyl)-1-piperazineethanesulfonic acid, Life Technologies), 2% (*v*/*v*) B27 supplement without vitamin A (Invitrogen, Carlsbad, CA, USA), 1% (*v*/*v*) N2 supplement (Invitrogen), 10 mM nicotinamide (Sigma-Aldrich, St Louis, MO, USA), 1.25 mM N-acetylcysteine (Sigma-Aldrich), 10% (*v*/*v*) R-spondin-1 conditioned medium (the Rspo1-Fc-expressing cell line was a kind gift from Calvin J. Kuo), 10 µM forskolin (Sigma-Aldrich), 5 µM A83-01 (transforming growth factor beta inhibitor; Tocris Bioscience, Bristol, UK), 50 ng/mL EGF (Invitrogen), 25 ng/mL HGF (Peprotech, Rocky Hill, NJ, USA), 0.1 µg/mL FGF10 (Peprotech) and 10 nM recombinant human (Leu15)-gastrin I (Sigma-Aldrich). Medium was changed twice a week. Passaging occurred every 7–10 days at ratios ranging between 1:2 and 1:4. All cultures were kept in a humidified atmosphere of 95% air and 5% CO_2_ at 37 °C. Organoids were primed for differentiation with BMP7 (25 ng/mL, Peprotech) through spiking EM 3 days before shifting to hepatic differentiation medium (DM). DM consisted of EM without R-spondin-1, FGF10, and nicotinamide, supplemented with 100 ng/mL FGF19 (Peprotech), 500 nM A83-01 (Tocris Bioscience), 10 µM DAPT (Selleckchem, Munich, Germany), 25 ng/mL BMP-7 (Peprotech), and 30 µM dexamethasone (Sigma-Aldrich). Organoids were kept on DM for up to 12 days. For exposure experiments, differentiation assay medium was prepared to reduce antioxidants in the medium. This assay medium consisted of DMEM GlutaMAX instead of DMEM-F12 medium complemented with the same components as differentiation medium excluding GlutaMAX, B27, and NAC. In 96-well format (cytotoxicity assay), 12,000 cells in 9 µL Matrigel per well were plated at the start of differentiation, and exposure was started at day 10 of differentiation for 48 h. In 24-well format (CYP activity assay), cells were densely plated in a fresh Matrigel droplet (50 µL) upon the start of differentiation, and cells were exposed to the CYP substrate cocktail on day 12 of differentiation. The median cell count after CYP cocktail assay was 290,415 (46,968–812,410).

All cell cultures were performed in a humidified atmosphere with 5% CO_2_ at 37 °C. The cellular morphology of the three cell models was visualized by immunofluorescence staining with a filamentous actin (F-actin) probe conjugated to a photostable green-fluorescent Alexa Fluor 488 dye (ThermoFisher) using confocal laser scanning microscopy (SP8, Leica Microsystems, the Netherlands), as described by Wang et al., 2022 [69].

### 4.2. Whole Genome RNA Sequencing

For mRNA sequencing, ICOs of three independent donors in expanding conditions and differentiated for 5 and 12 days were collected. Additionally, freshly isolated hepatocytes and liver tissue were used. RNA was isolated using the RNeasy Mini Kit (Qiagen, Hilden, Germany) according to the manufacturer’s instructions. As described in Schneeberger et al. (2018), Poly(A) Beads (NEXTflex, Bio Scientific, Austin, TX) were used to isolate the polyadenylated mRNA fraction. Sequencing libraries were prepared using the Rapid Directional RNA-Seq Kit (NEXTflex). Illumina NextSeq500 sequencing produced single-end 75-base-pair long reads. RNA-sequencing reads were mapped using STAR (v2.4.2a). Read groups were added to the BAM files with Picard’s AddOrReplaceReadGroups (v1.98) and sorted with Sambamba (v0.4.5). Transcript abundances were quantified with HTSeq-count (v0.6.1p1) using the union mode. The raw files were uploaded to Gene Expression Omnibus (GEO) database (accession number GSE123498). RNA sequencing data of the HepaRG cell line were retrieved from the GEO database (accession number GSE14654). Genes important in drug metabolism were selected [70]. Heatmaps were generated using edgeR.

### 4.3. Cytochrome P450 Activity

CYP activity was assessed by the addition of a CYP substrate cocktail prepared in assay medium (see methods cell culture). Two CYP cocktail sets were prepared to expose the hepatic cell models (Table 3): set A included phenacetin (CYP1A2, 15 µM), midazolam (CYP3A4, 5 µM), dextromethorphan (CYP2D6, 15 µM), tolbutamide (CYP2C9, 20 µM); and set B included 7-hydroxycoumarin (UGT, 12 µM), chlorzoxazone (CYP2E1, 25 µM) and bupropion (CYP2B6, 20 µM) [71,72]. At three cell model-specific timepoints (PHHs: 1, 2, 4 h; HepaRGs: 2, 4, 8 h; ICOs: 4, 8, 24 h), 400 µL exposure medium was placed into glass vials containing 400 µL acidified MeOH (0.1% (*v*/*v*) formic acid). The samples were stored at −20 °C until analysis. Prior to LC-MS/MS analysis, samples were centrifuged for 10 min at 1250 g to precipitate any protein.

Standards for LC-MS/MS analysis of phenacetin, acetaminophen, midazolam, hydroxy-midazolam, dextromethorphan, dextrorphan, tolbutamide, 4-hydroxy-tolbutamide, 7-hydroxy-coumarin, 7-hydroxy-coumarin glucuronide, chlorzoxazone, 6-hydroxy-chlorzoxazone, bupropion, and hydroxy-bupropion were prepared in the same matrix as the medium extracts. All chemicals were obtained from Sigma-Aldrich. All previously listed substrates and metabolites were analyzed in a single run using a Shimadzu triple-quadrupole LCMS 8050 system with two Nexera XR LC-20AD pumps, a Nexera XR SIL-20AC autosampler, a CTO-20AC column oven, an FCV-20AH2 valve unit (Shimadzu, ‘s Hertogenbosch, the Netherlands). The substrates and metabolites were separated on a Synergi Polar-RP column (150 × 2.0 mm, 4 µm, 80 Å) with a 4 × 2 mm C18 guard column (4 × 2 mm; Phenomenex, Torrance, CA, USA). The mobile phase consisted of 0.1% (*v*/*v*) formic acid in Millipore (A) and 0.1% (*v*/*v*) formic acid in MeOH (pH 2.7; B), and was set as 100% A (0–1 min), 100% to 5% A (1–8 min), 5% A (8–9 min), 5% to 100% A (9–9.5 min), and 100% A (9.5–12.5 min). The total run time was 12.5 min, and the flow rate was 0.2 mL/min. Peaks were integrated using LabSolutions software.

### 4.4. Cytotoxicity

Cells plated in 96-well plates were exposed to a concentration range (six concentrations in 2x dilution) of five known hepatotoxic compounds for 48 h (single dose). Acetaminophen (CAS 103-90-2; 30 mM) and valproic acid (CAS 1069-66-5; 10 mM) were directly dissolved in assay medium (previously described for each hepatic cell model). Diclofenac (CAS 15307-79-6; 100 mM), perhexiline (CAS 6724-53-4; 6.3 mM), and troglitazone (CAS 15307-79-6; TRC, Toronto, Canada; 40 mM) were dissolved in dimethyl sulfoxide (DMSO), which was 200x diluted in the highest exposure concentration. For the latter, vehicle control (0.5% (*v*/*v*) DMSO) was used in the exposure experiments. All chemicals were obtained from Sigma-Aldrich, unless stated otherwise. Dose ranges of exposure: acetaminophen 0.94–30 mM; diclofenac 15.57–500 µM; perhexiline 0.5–31.5 µM; troglitazone 6.25–200 µM; valproic acid 0.31–10 mM.

### 4.5. Cell Viability

The viability of exposed cells was determined by cellular ATP levels using the CellTiter-Glo Luminescent Cell Viability Assay (Promega, Madison, WI, USA). The CellTiter Glo reagent was prepared according to the manufacturer’s instructions. Briefly, the culture plate was equilibrated at room temperature for 30 min. Medium was removed from the plate, after which phosphate-buffered saline (PBS) and the CellTiter Glo reagent were added to each well in equal volumes. The plate was mixed for 2 min on an orbital shaker and incubated for an additional 10 min at room temperature. Luminescence was measured on the TriStar2 (Berthold Technologies, Bad Wildbad, Germany), and samples were normalized to (vehicle) control.

## 5. Conclusions

Here, we explored the levels of drug metabolism in hepatocyte-like intrahepatic cholangiocyte organoids and their potential in in vitro toxicity testing. We found that although the hepatic differentiation and the expression and activity of most drug metabolizing enzymes in ICOs were still below that of PHHs and HepaRGs, HL-ICOs could be a valuable platform for individualized toxicity screenings in the future. Further elucidation of enzyme and transporter activity in drug metabolism in HL-ICOs is needed to better define their potential. Additionally, exposure to a more extensive compound set including subtoxic concentrations and mechanistic studies would give more insight into the specific application of HL-ICOs in in vitro toxicity testing.

## Figures and Tables

**Figure 1 molecules-28-00621-f001:**
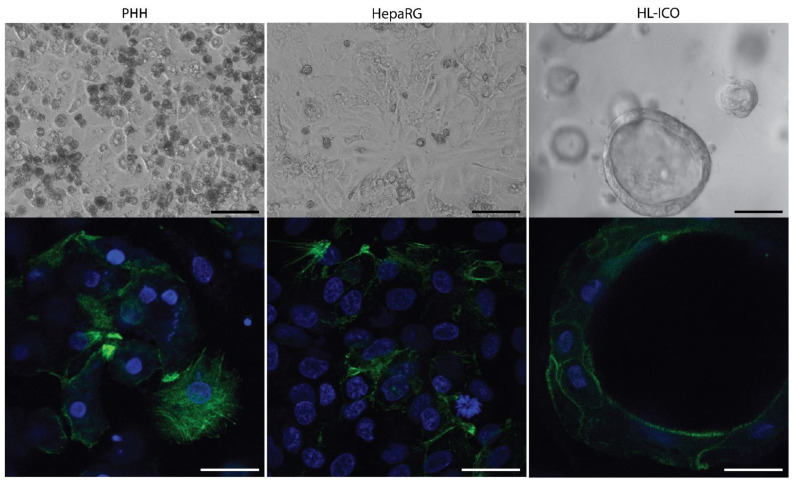
Morphology of primary human hepatocytes (PHHs), hepatic cell line HepaRG and hepatocyte-like intrahepatic cholangiocyte organoids (HL-ICOs) on differentiation day 12. Top: Brightfield pictures of morphology. Scale bar = 100 µm. Bottom: Phalloidin staining (green) of filamentous actin. Scale bar = 25 µm.

**Figure 2 molecules-28-00621-f002:**
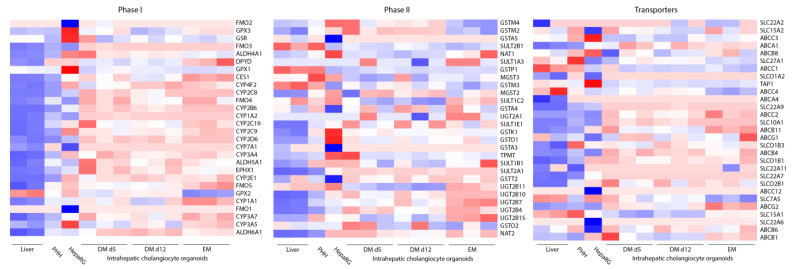
Heatmap showing the mRNA expression of intrahepatic cholangiocyte organoids from three independent donors in expansion (EM) and after hepatocyte differentiation (DM day 5 and day 12) compared with that of liver tissue, primary human hepatocytes (PHHs) and HepaRGs involved in drug metabolism (phase I (*n* = 28), phase II (*n* = 28), and transport (*n* = 29)). Red indicates low expression. Blue indicates high expression.

**Figure 3 molecules-28-00621-f003:**
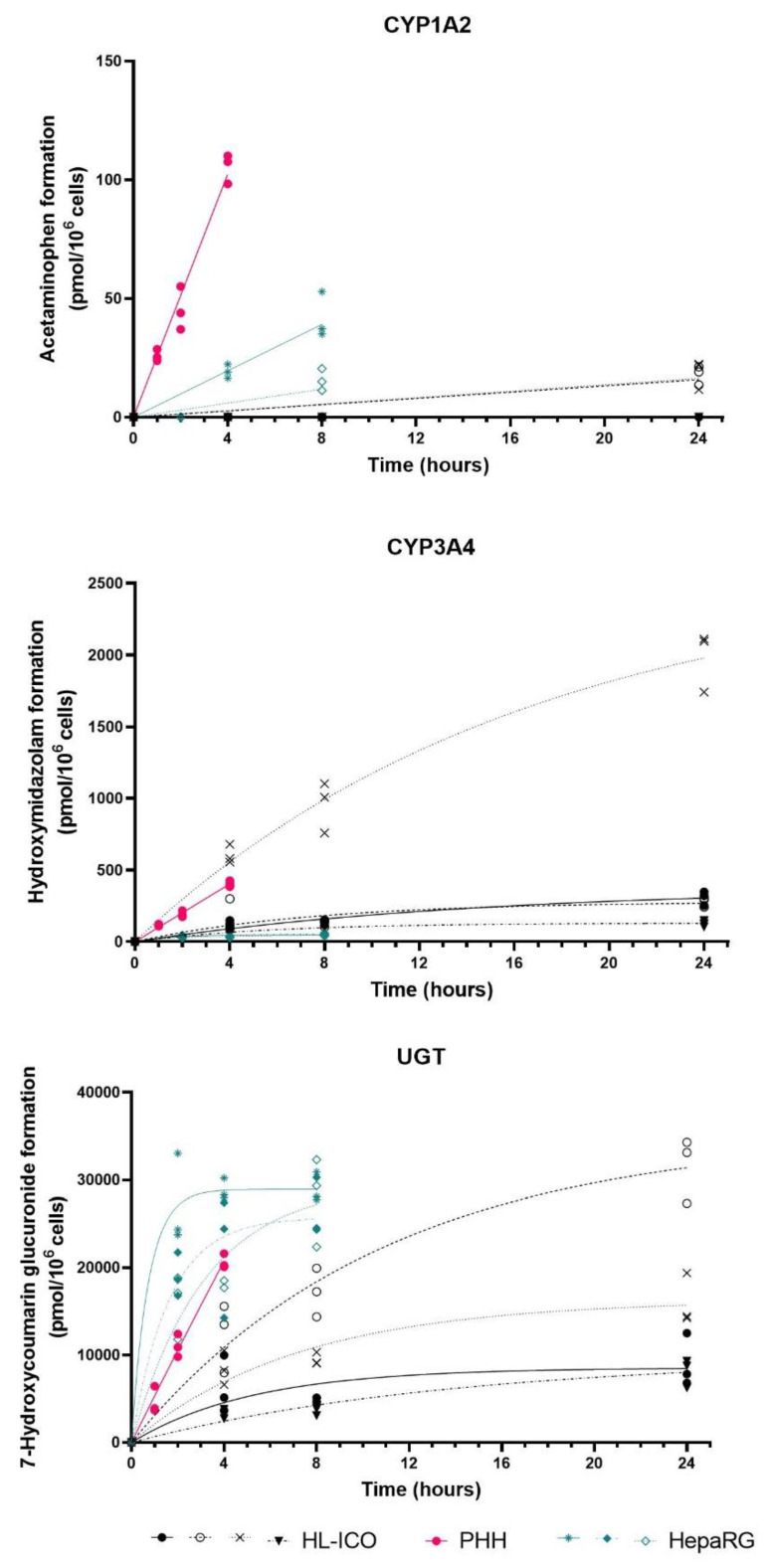
Metabolite formation as a measure of enzyme activity in hepatocyte-like intrahepatic cholangiocyte organoids (HL-ICOs) on differentiation day 12, primary human hepatocytes (PHHs), and HepaRG cells. HL-ICO: each black symbol indicates a different donor (*n* = 4). HepaRG: Each green symbol represents an independent experiment. PHH: Technical triplicates are shown in pink.

**Figure 4 molecules-28-00621-f004:**
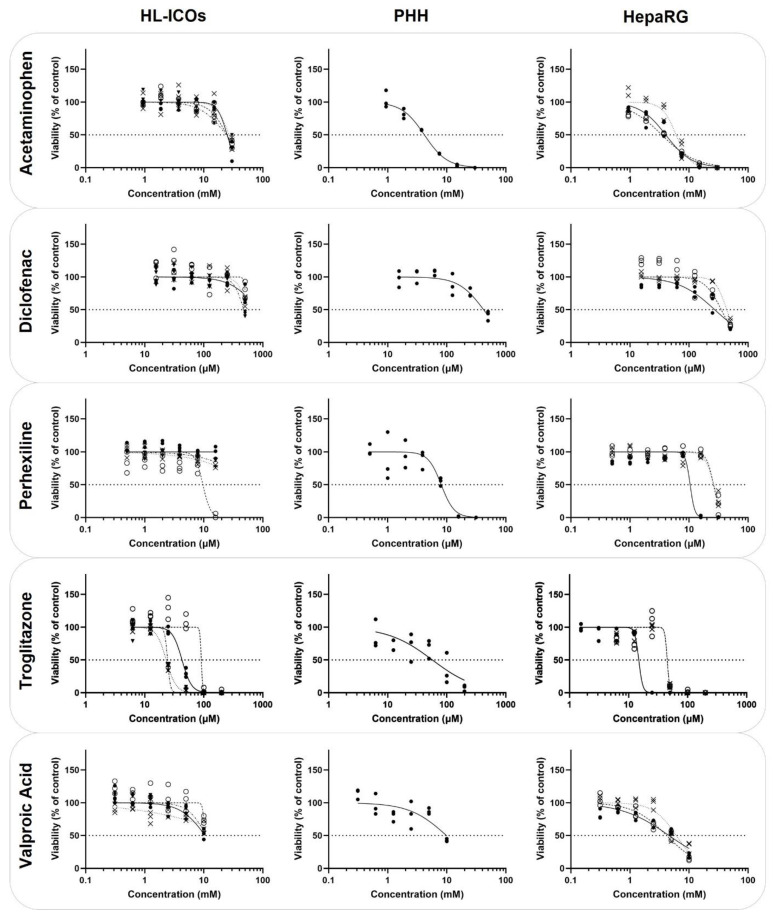
Sensitivity to known hepatotoxic compounds of HL-ICOs, PHHs, and HepaRGs. PHHs, HepaRG cells, and HL-ICOs (differentiation day 12) were exposed to acetaminophen, diclofenac, perhexiline, troglitazone, and valproic acid for 48 h (single dose). Data are presented as the percentage relative to the viability of (vehicle-treated) controls. For PHHs, one replicate experiment is shown. For HepaRG cells, three replicate experiments (different symbols) are shown (replicate experiments can be different between tested compounds). For HL-ICOs, four replicate experiments using different donors are shown, represented by different symbols. For all cell models, three replicate measurements per concentration are shown. Dashed line indicates 50% viability.

**Table 1 molecules-28-00621-t001:** Comparison of enzyme-specific metabolite formation rates in HL-ICOs, PHHs and HepaRG.

	HL-ICOs				PHH	HepaRG
CYP1A2	nd	0.01249	0.01293	nd	0.4269	0.03237 (nd–0.08552)
CYP2B6	nd	nd	nd	nd	50.13	1.228 (0.9913–1.358)
CYP2C9	nd	nd	nd	nd	15.33	2.161 (1.365–2.521)
CYP2D6	nd	nd	nd	nd	9.601	0.1656 (0.1567–0.2866)
CYP2E1	nd	nd	nd	nd	nd	nd
CYP3A4	0.3294	0.3603	2.098	0.2278	1.680	0.3136 (0.2645–0.3296)
UGT	13.18	38.93	22.89	8.778	86.79	158.9 (132.6–225.4)

Values presented are the metabolite formation rates (pmol/min/10^6^ cells). Hepatocyte-like ICOs (HL-ICOs; differentiation day 12): values are calculated per donor. PHH: Value represents the mean of a technical triplicate. HepaRG: Value represents median of three independent experiments, minimum and maximum formation rate within brackets. nd: not determinable (i.e., no metabolite formation).

**Table 2 molecules-28-00621-t002:** Determined EC50 values in HL-ICOs, PHH, and HepaRGs.

	HL-ICOs				PHH	HepaRG
Acetaminophen	24,870	24,630	26,840	24,010	4186	4036 (3465–6045)
Diclofenac	>500	>500	>500	475.5	421.2	351.7 (272.4–434.9)
Perhexiline	>31.5	9.675	>31.5	>31.5	8.072	25.97 (10.45–26.37)
Troglitazone	42.80	90.83	23.13	24.40	57.09	45.15 (14.89–45.17)
Valproic Acid	>10,000	>10,000	>10,000	>10,000	9885	4582 (4168–6066)

Values in µM. HepaRG cells: the median value of three independent experiments is shown with the minimum and maximum EC50 within brackets. HL-ICOs (differentiation day 12): determined EC50 values are shown for each donor separately.

**Table 3 molecules-28-00621-t003:** Information on enzyme activity cocktails.

	Enzyme	Parent Compound	CAS Number	Dosed Concentration (µM)
Cocktail A	CYP1A2	Phenacetin	62-44-2	15
		Acetaminophen	103-90-2	
	CYP3A4	Midazolam	59467-70-8	5
		Midazolam-OH	59468-90-5	
	CYP2D6	Dextromethorphan	125-71-3	15
		Dextrorphan	143-98-6	
	CYP2C9	Tolbutamide	64-77-7	20
		4OH-Tolbutamide	5719-85-7	
Cocktail B	UGT	7-OH Coumarin	93-35-6	12
		7-OH Coumarin Glucuronide	66695-14-5	
	CYP2E1	Chlorzoxazone	95-25-0	25
		6OH-Chlorzoxazone	1750-45-4	
	CYP2B6	Bupropion	31677-93-7	20
		OH-Bupropion	92264-81-8	

## Data Availability

Not applicable, except for RNAseq data: GEO accession number GSE14654.

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
