# Peer review of "Drug Metabolism of Hepatocyte-like Organoids and Their Applicability in In Vitro Toxicity Testing"

_molecules, 2023, doi:10.3390/molecules28020621_

Round 1

Reviewer 1 Report

The manuscript "Drug metabolism of hepatocyte-like organoids and their applicability in in vitro toxicity testing" explored the potential use of differentiated hepatocyte-like ICOs in in vitro toxicity testing defining the expression and activity of genes involved in drug metabolism. Eventhough the topic is emerging and very interesting some points remain to be adressed:

1) Authors  did non show any data about the characterization of obtained HL-ICOs and about he basal levels of the specific enzymes analyzed in the paper.

2) It can be interesting to analyze if the treatements can affect the transcriptomic profile of HL-ICOs and can allow to identify novel genes involved in toxicity

Author Response

We thank the reviewer for the thorough assessment of our manuscript. Please see the response in the attachment.

Reviewer 2 Report

The article: "Drug metabolism of hepatocyte-like organoids and their applicability in in vitro toxicity testing", written by Bouwmeester, provides useful and some new  information important for the field of in vitro toxicology. In my opinion, the manuscript is well conducted and the experiments were performed competently.

Minor points:

Abstract

1) I would suggest to rewrite the abstract and include some quantitative values in it. Abstract should be as quantitative as possible for rapid comparison with others studies, referring quantitative values.

It is written (Abstract, row 20): Here, we explored the potential of differentiated hepatocyte-like ICOs (HL-ICOs) in in vitro toxicity testing by exploring the expression and activity of genes involved in drug metabolism, a key determinant in drug-induced toxicity. Some of the presented results could be briefly reported here.

2) It is written (Abstract, row 28): Exposure to the hepatotoxicants showed comparable EC50s in HL-ICOs to PHHs and HepaRGs, only for acetaminophen exposure HL-ICOs were less sensitive. I would suggest to provide the obtained EC50 values.

General comments:

1) The abbreviation list of the tested drugs will be useful for readers.

2) The abbreviation of the tested drugs should be avoided in a subtitle (row 182)

3)  in vitro should be written in italic

Author Response

We thank the reviewer for the thorough assessment of our manuscript. Please see our response in the attachment.

Round 2

Reviewer 1 Report

No more comments. Authors answered to all questions.